# *Gastrodia elata* Blume Polysaccharides: A Review of Their Acquisition, Analysis, Modification, and Pharmacological Activities

**DOI:** 10.3390/molecules24132436

**Published:** 2019-07-02

**Authors:** Haodong Zhu, Chen Liu, Jinjun Hou, Huali Long, Bing Wang, De’an Guo, Min Lei, Wanying Wu

**Affiliations:** 1School of Pharmacy, Shanghai University of Traditional Chinese Medicine, Shanghai 201203, China; 2Shanghai Research Center for Modernization of Traditional Chinese Medicine, National Engineering Laboratory for TCM Standardization Technology, Shanghai Institute of Materia Medica, Chinese Academy of Sciences, Shanghai 201203, China; 3University of Chinese Academy of Sciences, Beijing 100049, China; 4School of Pharmacy, Nanchang University, Nanchang 330006, China

**Keywords:** *Gastrodia elata Bl* (*G. elata*), polysaccharides, phytochemistry, pharmacological action

## Abstract

*Gastrodia elata* Blume (*G. elata*) is a valuable Traditional Chinese Medicine (TCM) with a wide range of clinical applications. *G. elata* polysaccharides, as one of the main active ingredients of *G. elata*, have interesting extraction, purification, qualitative analysis, quantitative analysis, derivatization, and pharmacological activity aspects, yet a review of *G. elata* polysaccharides has not yet been published. Based on this, this article summarizes the progress of *G. elata* polysaccharides in terms of the above aspects to provide a basis for their further research and development.

## 1. Introduction

*Gastrodia elata* Blume (*G. elata*) is a precious traditional Chinese herbal medicine which was initially recorded in *Shen Nong’s Herbal Classic* about two thousand years ago. In the clinic, *G. elata* is used widely for the treatment of headaches, epilepsy, dizziness, rheumatism, neuralgia, cramps, high blood pressure, and other neurological diseases [1]. In the search for the active ingredients of *G. elata*, a series of small molecule compounds were found, including gastrodin, parishin, phenolic compounds, 4-hydroxybenzyl alcohol, and β-sitosterol [2,3,4,5,6]. Some of these small molecule compounds showed activity against headaches, high blood pressure [7], epilepsy, and other neurological diseases [2,5,6]. In 1981, Liu reported *G. elata* polysaccharides as another important active ingredient [8]. Since then, *G. elata* polysaccharides have received wide attention throughout the world, and a series of pharmacological activities were reported, such as anticancer [9], antivirus [10], antiosteoporosis [11], antioxidant [12], immunomodulatory [13], and neuroprotective effects [14], as well as a great effect on the cardiovascular system [15]. These results clearly indicate that *G. elata* polysaccharides play a key role in *G. elata’s* pharmacological activities.

Due to the importance of *G. elata* polysaccharides, it is necessary and urgent to summarize their actions. Up to now, studies on *G. elata* polysaccharides have focused on their extraction, purification, qualitative/quantitative analysis, derivatization, and pharmacological activity. However, no previous articles have synthetically summarized the research papers on *G. elata* polysaccharides. In this article, we summarize and review all the mentioned aspects of *G. elata* polysaccharides. The purpose of this article is to contribute to further and deeper research on the extraction, analysis technologies, quality control, bioactivities, and derivatization of *G. elata* polysaccharides.

## 2. Acquisition of *G. elata* Polysaccharides

### 2.1. Extraction of G. elata Polysaccharides

Phytomedicines contain fat-soluble ingredients which may interfere with the extraction of polysaccharides. Therefore, before the extraction of polysaccharides degreasing using organic solvents, including chloroform–ethanol [16,17], ether [17], petroleum ether [18,19], methanol [20], and different concentrations of ethanol [20,21] is required to remove some interfering components [22]. *G. elata* can be degreased with different concentrations of ethanol, such as 75% ethanol [23], 80% ethanol [24], 85% ethanol [25,26], and 95% ethanol [10,27]. Chen et al. obtained crude *G. elata* polysaccharide (RGP) in a yield of 6.11% after degreasing with 75% ethanol [23]. Lee et al. obtained 2.47 g of crude *G. elata* polysaccharide degreased with 80% ethanol [24]. Ming et al. found that the yield of *G. elata* polysaccharide (PGEB-3*H*) degreased with 85% ethanol was 0.797 g/kg [25]. Chen et al. found that the yield of crude *G. elata* polysaccharide (TM) degreased with 95% ethanol was 5.2% [27]. The relationship between the yield and degreasing concentration is vague and may be related to other extraction factors, but these results show that ethanol solution is a suitable solvent for the removal of fat-soluble ingredients from *G. elata*.

Water is widely used as the solvent to extract *G. elata* polysaccharides under different conditions. Most of the *G. elata* polysaccharides, e.g., glucan PGEB-3*H* [25,26], WGEW [10], WTMA [27], RGP-1a and RGP-1b [23], the acidic polysaccharides [24] and PGE [28], were extracted by traditional heating methods with different extraction times, extraction frequencies (for assisted methods), and extraction temperatures. Specific information can be found in Table 1. Qiu et al. obtained the alkali-soluble polysaccharide (AGEW) by extraction with 5% NaOH at 4 °C for 2 h [10]. Microwaves can penetrate the cell walls of Chinese medicines and interact with polar components to accelerate extraction [29]. Zhu et al. found the best method of extraction of *G. elata* polysaccharides, with a yield of 5.42%, to be microwave extraction with water (40×) at 120 °C for 3 h [30]. Ultrasound can enhance the conduction between plants and solvents [31,32] and destroy cell walls to improve the extractability of polysaccharides [33,34]. Zhang et al. indicated that the extraction rate of polysaccharides extracted with 45 volumes (*v*/*w*) of water at 66 °C for 34 min was 32.78% [35]. In addition the above methods, enzyme-assisted treatments, which can promote the release of polysaccharides, have also been applied to extract polysaccharides [36,37]. Tan et al. adopted the orthogonal method to determine the optimal process for enzymatic extraction of *G. elata* polysaccharides, and the amount of *G. elata* polysaccharides extracted was 46.63 mg/g under those conditions, involving an enzyme dosage of 8 mg/g [38]. Enzyme-assisted extractions are rarely used alone and are often combined with other extraction methods to increase the polysaccharide yield [39]. Wang et al. showed that the yield of *G. elata* polysaccharides extracted by hot water extraction was 22.380%, the yield of *G. elata* polysaccharides extracted by ultrasonic-assisted extraction was 33.089%, and the yield of *G. elata* polysaccharides enzymatically extracted was 50.315% [40]. At the same time, Wang et al. determined that the antioxidant activity of *G. elata* polysaccharides extracted by ultrasound in 2,2-diphenyl-1-picrylhydrazyl (DPPH) and ferric reducing antioxidant power (FRAP) antioxidant evaluation systems was higher than after enzymatic extraction and hot water extraction, and in the 3-ethyl-benzothiazoline-6-sulfonic acid (ABTS) antioxidant evaluation system, the activity of *G. elata* polysaccharides obtained by enzymatic extraction was higher than by ultrasound auxiliary extraction and hot water extraction [40]. On the whole, ultrasonic-assisted extraction and enzymatic extraction may benefit the extraction of *G. elata* polysaccharides while maintaining high antioxidant activity.

The extraction methods of *G. elata* polysaccharides are very traditional, and they have many drawbacks. First, a study showed that the yields of extraction by hot water are influenced by the extraction time, temperature, feed–liquid ratio, and other factors [41,42,43]. Long-term and high-temperature hot water extraction leads to degradation of the polysaccharides, which results in a decrease in the yield of *G. elata* polysaccharides. Second, although irradiation-assisted extraction can shorten the extraction time and improve the efficiency, microwaves [44,45] and ultrasound [46,47,48] can lead to the decomposition of *G. elata* polysaccharides. Third, the specificity and selectivity of enzymes are also influenced by temperature and pH. In recent years, some new technologies have been developed for polysaccharide extraction to improve the extraction efficiency, such as supercritical fluid extraction (SFE) [49], pressurized liquid extraction [50], and induced electric field [51]. SFE is an efficient, safe, and environmentally friendly method, which has been applied successfully to extract Pachyman from *Poriacocos (Schw.) wolf* [52]. Furthermore, a new method for extracting and isolating polysaccharides from *Semen cassiae* was established by microwave-assisted aqueous two-phase extraction [53]. The ethanol-soluble polysaccharides and water-soluble polysaccharides are separated simultaneously. These new extraction techniques could be used for the extraction of *G. elata* polysaccharides in order to obtain new polysaccharides with novel structures for research.

### 2.2. Purification of G. elata Polysaccharides

Aqueous extracts contain a lot of impurities, and protein removal is an important step in the purification process. Several methods are commonly used in deproteinization, such as the Sevag method, the trichlorotrifluoroethane method, and the trichloroacetic acid method. The merits of these methods are their compatibility with polysaccharides. Due to the fact the Sevag method is much milder than the others, *G. elata* aqueous extract is typically deproteinized using the Sevag method [23,41,54]. The enzyme method is another effective and alternative to the common methods to eliminate proteins. Zhu et al. found that the protein removal rate was 90.1% under an enzyme dosage of 2.0 U/mg for 5 h [55]. The use of proteolytic enzymes to remove proteins is an environmentally friendly method with a mild response. After the protein has been removed, small molecule impurities are removed by dialysis, and then ethanol is added to the dialysate for precipitation to obtain crude polysaccharides. After these steps, the crude polysaccharides are further separated to obtain pure polysaccharides.

Polysaccharides can be separated by column chromatography to obtain homogenous polysaccharides, for example, using ion-exchange chromatography and size exclusion chromatography. Typically, ion-exchange chromatography is used for the separation of neutral/acidic polysaccharides from negatively charged polysaccharides by gradient salt elution or pH changes [56]. The *G. elata* polysaccharides WGEW, AGEW, WTMA, and GPs are prepared by using DEAE-cellulose column chromatography to further purify the crude polysaccharides [10,13,27]. Size exclusion chromatography is usually used for separating polysaccharides according to differences in molecular weight or molecular size. For example, Zhu et al. separated the crude polysaccharides on a Sephadex G-200 column at a flow rate of 0.30 mL/min, and then collected and further purified them by ultra-filtration tubes to obtain G. *elata* polysaccharides (PGE) [28]. Moreover, it has been reported that polysaccharides, such as PGEB-3H [25,26], RGP-1a and RGP-1b, can be purified by ion-exchange chromatography combined with size exclusion chromatography [23]. The macroporous resin D101 purifies *G. elata* polysaccharides well and can increase the purity of *G. elata* polysaccharides to 65.7% under optimal purification conditions [57]. All of the above information on *G. elata* polysaccharides is listed in Table 2.

Size exclusion chromatography has the obvious drawback that it can only be used for the analysis of relatively high molar mass samples and degrades ultra-high molar mass polymers, which makes it impossible to judge the exact molar mass of the sample [22]. Asymmetrical flow field-flow fractionation (AF4) is an effective fractionation technique that can solve this problem. Unlike chromatography, AF4 uses a separation channel without fillers, rather than a packed column. It can be applied to the characterization of colloids and macromolecules using low pressure and mild processing [58]. In addition, high-speed countercurrent chromatography (HSCCC) has been widely applied for the separation of natural products [59]. As a new technique, HSCCC has been applied successfully to separate and purify polysaccharides [60,61,62,63,64]. These methods could be used in the separation of *G. elata* polysaccharides to obtain higher purity *G. elata* polysaccharides for research.

## 3. Analysis of *G. elata* Polysaccharides

### 3.1. Qualitative Analysis of G. elata Polysaccharides

Analysis of the monosaccharide composition, ratio, and glycosidic linkages of polysaccharides is the most important step in the analysis of polysaccharides. The *G. elata* polysaccharides are degraded to monosaccharides by acid hydrolysis, and then these monosaccharides are separated and analyzed by various chromatographic techniques. The main advantages of thin layer chromatography (TLC) are its easy sample operation and the low cost required to detect the monosaccharide composition. Using TLC, Lee et al. found that the acidic polysaccharides obtained from *G. elata* included xylose, glucose, galacturonic acid, and glucuronic acid [24]. Gas chromatography (GC) is the most common method used to detect monosaccharide composition. Derivatization of the monosaccharides, for example, using methylation and acetylation, is required prior to the use of gas chromatography. Previous studies using GC analysis, indicated that *G. elata* polysaccharides are mainly composed of the monosaccharide glucose [10,13,25,27,28]. High performance liquid chromatography is an important method for the detection of polysaccharides. Using a refractive index detector, Chen et al. reported that *G. elata* RGP-1a is composed of fructose and glucose in a ratio of 1:10.68, and *G. elata* RGP-1b is only composed of glucose [24]. Using a carbohydrate column and a pulsed amperometric detector, Zhu et al. reported that the *G. elata* polysaccharide GEP is composed of glucose [54]. Through the use of ion chromatography to analyze the monosaccharide composition of *G. elata* polysaccharides, Li et al. reported that the *G. elata* polysaccharide is composed of rhamnose, galactose, glucose, xylose, and mannose [65].

Traditional acid hydrolysis will lead to a range of hydrolyzed monosaccharide ratios due to the poor selectivity of acid hydrolysis. To overcome this drawback, the use of specific glycosidases to digest polysaccharides has gradually replaced traditional acid hydrolysis methods. After digestion of the polysaccharides, the resulting monosaccharides can be separated by chromatographic techniques, such as high performance thin layer chromatography, high performance liquid chromatography, capillary electrophoresis tubes, PACE, and so on. Saccharides without ultraviolet absorption cannot be detected with an ultraviolet detector, unless the saccharide is derivatized. The evaporative light-scattering detector (ELSD) [66] and charged aerosol detector [67] can directly detect polysaccharides without derivatization. Matrix-assisted laser desorption/ionization can identify most polymers without derivatization and has been successfully used for the characterization of polysaccharides, such as seaweed polysaccharides [68]. Characterization and qualitative analysis are performed by the above methods [68,69,70]. Carbohydrate gel electrophoresis (PACE) has been applied to study the polysaccharides in *Cordyceps* [71,72,73] and *Ganoderma* [74]. Due to the individual preparation of the gel, the reproducibility of the glycan profile should be considered. High-efficiency thin-layer chromatography does not have this issue and can characterize ginseng polysaccharides [75]. These new techniques for separating monosacccharides after polysaccharide digestion could be applied for the analysis of *G. elata* polysaccharides to improve the accuracy of the analysis of *G. elata* polysaccharides.

Mass spectrometry is an efficient analysis method to obtain molecular information on polysaccharides. It is commonly used in the structural analysis of polysaccharides in conjunction with nuclear magnetic resonance (NMR) spectroscopy. NMR data (1D and 2D NMR) provide information on aspects of polysaccharide structure, such as the monosaccharide composition, sugar configuration, connection characteristics, and monosaccharide connection sequence. Some *G. elata* polysaccharides have a repeating structure with an α-(1→4) glucan and an α-(1→4) branch at O-6 according to NMR and GC-MS [10,25,27]. However, the structure of the polysaccharide PGE obtained from *G. elata* is a backbone of (1→4)-linked-d-Glcp and 1→3 and 1→4,6-branched glucopyranose [28]. Zhou et al. isolated the polysaccharide GEP II from *G. elata*, which was composed of glucose and mannose [76]. The main skeleton has 1→6 and 1→4 glycosidic bonds as well as 1→2 glycosidic bonds [76]. The monosaccharide composition, backbone, and molecular weight of polysaccharides derived from *G. elata* are summarized in Table 3.

### 3.2. Quantitative Analysis of G. elata Polysaccharides

The extraction and purification of polysaccharides are key steps to obtain polysaccharides, which have a great influence on the polysaccharide composition and content. At present, the quantitative analysis methods for polysaccharides mainly include colorimetry, high performance liquid chromatography, and gas chromatography. The total content of *G. elata* polysaccharides is usually determined by colorimetric methods of different color systems, such as the phenol–sulfuric acid method and the anthrone–sulfuric acid method. Because of the differences between monosaccharides, when glucose is used as a reference, the monosaccharide composition has a great influence on the polysaccharide content [77]. The monosaccharide content of a polysaccharide can be quantified by GC and HPLC to calculate the number of monosaccharides released after hydrolysis of the polysaccharide. For example, using HPLC with an RI detector, Chen et al. found that *G. elata* RGP-1a is composed of fructose and glucose in a ratio of 1:10.68 [23]. High performance gel permeation chromatography (HPGPC) is ofter applied to determine the molecular weights of polysaccharides. A series of glucans with different molecular weights is used to establish the standard curve and determine the molecular weights of polysaccharides. All the molecular weights of *G. elata* polysaccharides, such as the polysaccharides WGEW, AGEW [10], WTMA [27], PGEB-3H [25], RGP-1a, RGP-1b [23], GPs [13], PGE [28], and GEP [54], are determined according to this method.

Application of both the colorimetric method and the HPGPC method for *G. elata* are too complicated, as they require an individual polysaccharide reference. Natural polysaccharides can be isolated by HPSEC, and then the molecular mass of each component can be determined by multi-angle laser light scattering. The polysaccharide or polysaccharide component can be quantified according to the response of the polysaccharide to the refractive index detector and the refractive index increment (dn/dc). The method without a reference substance is simple and accurate. The method has been used to quantitatively analyze the polysaccharides for species in the *Panax* genus, e.g., *P. ginseng*, *P. notoginseng,* and *P. quinquefolius* [78], as well as *Lycium barbarum* [79]. 

Previous studies suggest that several specific polysaccharides from *G. elata* with defined structures clearly consist only of glucose and have a main chain of 1–4 glycosidic bonds, for example, AGEW, WGEW, WTMA and PGEB-3*H* [10,25,27]. The structures of several polysaccharides are shown in Figure 1. Some polysaccharides from *G. elata* without a defined structure consist of glucose and several other monosaccharides. The similar structures of some *G. elata* polysaccharides and its monotonous monosaccharide compositions may be related to the lack of novelty of the research method of *G. elata* polysaccharides. Therefore, the development of a new research method for *G. elata* polysaccharides may result in a novel structure of *G. elata* polysaccharides, which, in turn, has promoted research on *G. elata* polysaccharides.

## 4. Modification of *G. elata* Polysaccharides

Polysaccharides from traditional Chinese herbal medicines have received a lot of attention due to their wide range of biological activities. The structure–activity relationships of polysaccharides are complex and hard to determine and are thus a big obstacle to the advancement of polysaccharide research. The pharmacological effects of polysaccharides depend on their characteristics, including molecular weight, solubility, viscosity and other physical or chemical properties [80,81,82,83]. The molecular derivatization of polysaccharides can change and enrich the structure and physico-chemical properties of polysaccharides. Physical, chemical, and biological pathways are three major methods of polysaccharide molecular derivatization [84]. In addition, through modification and derivatization of the chemical structure of polysaccharides, qualitative or quantitative analyses of polysaccharides have been performed [22]. Hence, more and more research has focused on the derivatization and modification of polysaccharides [85].

Up until now, several methods for the modification of polysaccharides have been reported, including sulfation [86], phosphorylation [87], carboxymethylation [88], selenization [89], acetylation [90], acid/alkali degradation [91], and others [92,93,94]. In 2007, Ding and their collaborators first isolated two polysaccharides from *G. elata*, WGEW and AGEW, and found that these two polysaccharides only consist of glucose [10]. Based on the reported structure–activity relationship between sulfated polysaccharides and anti-dengue virus bioactivities, they then sulfated WGEW and AGEW [10]. Using the chlorosulfonic acid-pyridine method, two sulfated derivatives of two polysaccharides—WSS25, WSS45, ASS25, and ASS45—were obtained at 25 and 45 °C, respectively. The basic information and structures of WGEW and AGEW and their sulfated derivatives is shown in Table 4 and Figure 2 [10]. The antiviral activity of WSS25, WSS45, and ASS45 was shown to be stronger than that of WGEW and AGEW. However, the selectivity index (CC_50_/EC_50_) of virus inhibition activities of WSS25 and ASS45 was lower than that of WSS45. WSS45 was shown to be a better dengue virus type 2 (DV2) inhibitor, with a selectivity index of more than 1000 [10]. Further study on the antiviral mechanism of WSS45 revealed that WSS45 does not directly kill the virus. WSS45 mainly interferes with the adsorption of DV2 by target cells, thereby strongly inhibiting DV2 infection. Furthermore, they found that the antiviral effect is related to the molecular weight of WSS45 and the degree of sulfation (DS) [10]. It has also been reported that the antivirus ability of sulfated polysaccharides increases as the degree of sulfation increases [10]. We can see that there is no correlation between the molecular weight of different sulfurized polysaccharides and their final efficacy strength and selectivity index for DV2. The structure–activity relationship between sulfated polysaccharides with different DS or molecular weights and their antiviral activities could be inferred.

At the same time, the antiviral activity structure-activity relationship of sulfated polysaccharides is very complex. It is not enough to depend on only four sulfated polysaccharides to determine this structure–activity relationship. If the structure–activity relationship of sulfated polysaccharides from *G. elata* is to be studied further, more derivatization and pharmacological research on sulfated polysaccharides is essential.

According to the structure–activity relationship between anti-angiogenic target proteins (Id1 and HS) and sulfated polysaccharides, WSS25 was confirmed to inhibit Id1 expression in HMEC-1 cells and block the BMP2/Smad/Id1 signaling pathway (Figure 3), thereby reducing tumor angiogenesis and inhibiting hepatic tumor cells [95]. Another molecular mechanism by which WSS25 inhibits angiogenesis is the inhibition of HMEC-1 cells by inhibiting dicer, a key enzyme in miRNA biosynthesis [96].

Although the anti-angiogenic mechanism of WSS25 has been studied clearly, its structure–activity relationship is still complicated. Therefore, in order to further study the structure–activity relationship of the sulfated polysaccharides of WSS25 on its anti-angiogenesis effects [97], a dozen WSS25-based sulfated derivatives and several WGEW aminopropylation, carboxymethylation, phosphorylation, and acetylation derivatives were synthesized. Basic information about these derivatives is shown in Table 3. Only sulfated polysaccharides with a molecular weight higher than 41,000 Da were shown to have an anti-angiogenic effect. Because the longer sugar chain structure could interact with the target protein, polysaccharide derivatives containing other groups do not have this activity. The strength of anti-angiogenic activity depends on the DS of polysaccharide derivatives. Additionally, the derivatized polysaccharides with a DS value of 0.173 to 0.194 showed better activity. The glucose branch of the sulfated polysaccharide was found to contribute little to this activity.

It was also reported that WSS25 can interact with BMP-2 in hepatic cancer cells [95]. Meanwhile, BMP-2 plays an important regulatory role in osteoclast and osteoblast cells [98,99]. WSS25 can inhibit bone loss induced by ovariectomy in female mice. This result suggests that WSS25 might be an effective anti-osteoporosis drug for older women [11].

In summary, research on *G. elata* polysaccharide derivatives has mainly focused on the sulfated products WSS25 and WSS45. They have antiviral, antiangiogenic, and antiosteoporosis biological activity. Some regular patterns were discovered from research on the antiangiogenic structure-activity relationship of *G. elata* sulfate polysaccharides. Unfortunately, studies on other derivatives of *G. elata* polysaccharides, their related activities, and structure-activity relationships are rarely reported.

## 5. Pharmacological Activities and Functions of *G. elata* Polysaccharides

A lot of references point out that *G. elata* shows significant pharmacological activity [100,101]. The activities of *G. elata* polysaccharides are summarized in Table 5.

Research on the pharmacological activities of *G. elata* polysaccharides mentioned in this paper mainly involves two parts: animal experiments in vivo and cell experiments in vitro. While Liu et al. used intracortical injection as an administration method in their experiments [12], almost all animal experiments were performed by oral administration. However, none of the above studies on systemic activity mention any pharmacokinetic parameters and characteristics other than the mode of administration and the dosage of *G. elata* polysaccharides. The in vivo pharmacokinetic study of *G. elata* polysaccharides is still blank, and its molecular mechanism still needs further confirmation. The in vitro cell experiments mentioned mainly use a method in which the cells are co-incubated with the *G. elata* polysaccharides. Unfortunately, few studies have mentioned whether *G. elata* polysaccharides can enter cells or whether certain pathway affect cells.

### 5.1. Anti-Cancer Activities

Chen et al. reported that the growth of PANC-1 cells could be inhibited by WTMA, and the polysaccharides showed insignificant toxicity on PANC-1 cells but no inhibition effect on live LO2 cells [27]. Liu et al. found that *G. elata* polysaccharides have a significant anti-tumor effect on H22 tumor-bearing mice, which can reduce the tumor weight [9]. The high-dose *G. elata* polysaccharide inhibition rate was shown to reach 44.7%. A study showed that *G. elata* polysaccharide could increase the G0/G1 phase cell percentage and decrease the G2/M phase cell percentage. The mechanism of action might be related to cell cycle distribution, the suppression of cell proliferation, and the activation of the caspase system to induce tumor cell apoptosis [9].

### 5.2. Antioxidation Activities

*G. elata* polysaccharides show antioxidation activities [12]. Previous studies have revealed that *G. elata* polysaccharides can improve the activity of SOD in the brain and GSH-Px in the blood, and *G. elata* polysaccharides can inhibit MAO activity in the brain and reduce the level of MDA in the brain tissue of aging mice. Xie et al. reported that *G. elata* polysaccharides can improve the learning and memory ability of D-galactose–induced aging mice to improve the activity of enzymes related to oxidative metabolism in the body [102]. *G. elata* polysaccharides can also delay the aging of the human body related to free radicals and increase the activity of superoxide dismutase and glutathione peroxidase, as well as the serum malondialdehyde levels in aging mice through dose-dependent enhancement [52].

### 5.3. Immunological Activities

Chen et al. reported that RGP-1a and RGP-1b, isolated from *G. elata*, could enhance the NO production and phagocytic activity of RAW 264.7 macrophages in a dose-dependent manner [23]. Li et al. reported that for mice in a cyclophosphamide-induced immunocompromised state, the serum of IgA, IgG, and serum hemolysin were increased in the middle- and high-dose groups of *G. elata* polysaccharides (*p* < 0.01). The high-dose group increased the spleen index and thymus index, and the polysaccharide middle dose group significantly increased serum IgM levels (*p* < 0.05) [103]. The *G. elata* polysaccharide could alleviate the inhibitory effect of cyclophosphamide on humoral immune function in mice. Bao et al. showed that *G. elata* polysaccharides isolated from the dried rhizomes of *G. elata* augment serum IL-2, TNF-α, IFN-γ, IgG, IgA and IgM levels as well as the spleen and thymus indexes of Kunming mice with immunomodulatory activity in a dose-dependent manner following intragastric treatment [13].

### 5.4. Neuroprotection Activities

The research results show that the polysaccharide PGEB-3*H* can improve the learning and memory ability of mice with scopolamine-induced memory disorders through increasing the Ach content in brain tissue [14]. *G. elata* polysaccharides (PGB) were shown to increase the number of BDNF-positive cells and SCF-positive cells and decrease the average gray value, suggesting that PGB has neuroprotective effects by up-regulating BDNF and SCF expression in brain tissues around ischemic lesions [104]. Electroacupuncture combined with PGB may improve the neurological function of rats with focal cerebral ischemia rats by increasing Nestin and brain-derived neurotrophic factor expression to accelerate the growth of neural stem cells in the basolateral amygdala [107]. The PC12 cells were protected from corticosterone-induced apoptosis and lactate dehydrogenase leakage, and intracellular reactive oxygen levels were reduced after treatment with 1000 μg/mL of polysaccharides from *G. elata* (GEP) before exposure to 200 μM of corticosterone by suppressing the endoplasmic reticulum stress-mediated pathway [105].

### 5.5. Cardiovascular System Activities

The *G. elata* polysaccharide PGB was shown to have a good antihypertensive effect on RHR rats by promoting the production of endogenous vasoactive substances, such as nitric oxide, and inhibiting the release of endogenous vasoconstrictors, such as plasma endothelin and angiotensin II [15]. By feeding PGEB-3*H* to rats with hyperlipidemia, Ming et al. showed that the polysaccharide PGEB-3*H* has potential lipid-lowering effects [25]. The crude and acidic polysaccharides of *G. elata* were shown to suppress the total cholesterol and LDL concentrations to decrease the atherosclerosis risk, and there were no significant differences in the serum triglyceride and HDL levels in SD rats fed a high-fat diet, indicating that the acidic polysaccharide of *G. elata* might significantly reduce the risk of cardiovascular disease (CVD) and atherosclerosis through suppressing the de novo synthesis of total cholesterol and LDL [106]. The acidic polysaccharides isolated from *G. elata* were shown to decrease blood pressure and improved serum lipid levels in SHR rats fed a high-fat diet by reducing the total cholesterol, triglyceride and LDL levels [24].

### 5.6. Other Functions

In addition to their good pharmaceutical potential, *G. elata* polysaccharides can also be applied in cosmetics and foods. *G. elata* polysaccharides show good functions related to appropriate viscosity, moisture absorption, and moisture retention due to their numerous hydroxyl groups [108]. *G. elata* polysaccharides also have good antioxidant and anti-aging activities [12,52]. A series of studies have applied *G. elata* polysaccharides to the preparation and development of moisturizers and skin creams [109,110,111]. Based on the anti-hypertension [15], anti-cancer [9,27], and anti-aging activities of water-soluble *G. elata* polysaccharides [12,52], Zhengjie and colleagues developed a *G. elata* polysaccharide-functional beverage [111].

## 6. Discussion

At present, the extraction of *G. elata* polysaccharides is based on traditional methods, including hot water extraction, ultrasonic extraction, microwave extraction, and so on. All of these methods use water as the extraction solvent and organic solvents for the removal of low-polarity small molecule compounds. The existing greener extraction methods, such as the microwave-assisted aqueous two-phase method [53], SFE extraction [52], and ionic liquids [112], could be applied to extract *G. elata* polysaccharides. Ionic liquids consist of organic cations and organic or inorganic anions, which exist in a liquid state at room temperature [112]. Ionic liquids have good solubility for bioactive polysaccharides, because of the stronger hydrogen bonds resulting from the interaction between polysaccharides and the large number of anions in the ionic liquid (such as AcO^−^ and Cl^−^) [111]. For example, chitosan and β-d-glucan both exhibit good solubility in 1-ethyl-3-methylimidazolium acetate (EMIMAc) [113,114]. Therefore, we can infer that SFE and ionic liquids with melting points in the room temperature range and can be used as novel and green non-aqueous extraction solvents for the extraction of *G. elata* polysaccharides.

So far, the structures of only a few polysaccharides isolated from *G. elata* have been defined clearly, such as WTMA [27], WGEM [10], AGEM [10], and PGBE-3H [25]. Polysaccharides without a characterized specific structure do not have assured reproducibility, which is not conducive to further ensuring their clinical efficacy and drug stability. The structure–functional relationships of *G. elata* polysaccharides represent a blank field that needs further research to control the quality of polysaccharides. Wu et al. used the saccharide mapping method to control the quality of polysaccharides from *Ganoderma spp.* [74]. Saccharide mapping is a new quality control method for uncharacterized polysaccharides. It is based on a specific and mild digestion method, followed by chromatographic separation including HPTLC, HPSEC, CE, and PACE [21,70], which does not require knowledge of the clear structure of polysaccharides. However, saccharide mapping has not been put into use for the quality control of *G. elata* polysaccharides; it is necessary to develop a simple and reliable method for the quality control of *G. elata* polysaccharides to guarantee the quality of drugs developed using *G. elata* polysaccharides.

It is undeniable that *G. elata* polysaccharides have good biological activity in many aspects, including anticancer, antioxidation, antihypertensive, immune system modulation and nervous system regulation. Especially, the function of *G. elata* polysaccharides in neuroprotection and treatment of hypertension diseases has great potential to improve the health of middle-aged and elderly populations. However, most of the current studies on systemic pharmacological activity have neglected the mechanism of action of *G. elata* polysaccharides in animal experiments in vivo. The molecular mechanisms have also not been fully investigated in in vitro cell experiments. The field of *G. elata* polysaccharide pharmacokinetic studies is still blank.

In the derivatization of *G. elata* polysaccharides, research has mainly focused on sulfurized polysaccharides, resulting in a lack of information about other derivatives of *G. elata* polysaccharides. The anti-angiogenic biological activity of WSS25 and its structure–activity relationship have been well researched. However, the bioactivity and structure–activity relationship of other derivatized compounds of *G. elata* polysaccharides are still unknown. Based on the structure–activity relationships of other derivatized polysaccharides, there is potential to study the preparation, pharmacological activity, and structure–activity relationships of *G. elata* polysaccharides. However, the difficulty lies in the fact that the uniform composition and complex structure of *G. elata* polysaccharides causes great disturbance to the derivatization.

## 7. Conclusions

In summary, many new ideas and techniques have been developed for the study of polysaccharides in traditional Chinese medicine. However, the extraction, analysis, and derivatization of *G. elata* polysaccharides are still conducted using traditional methods. Consequently, *G. elata* polysaccharides have not been fully studied. There is still much room for deeper research. New extraction, analysis, and derivatization research methods should be applied to the study of *G. elata* polysaccharides to further discover its potential bioactivity, functions, and applications.

## Figures and Tables

**Figure 1 molecules-24-02436-f001:**
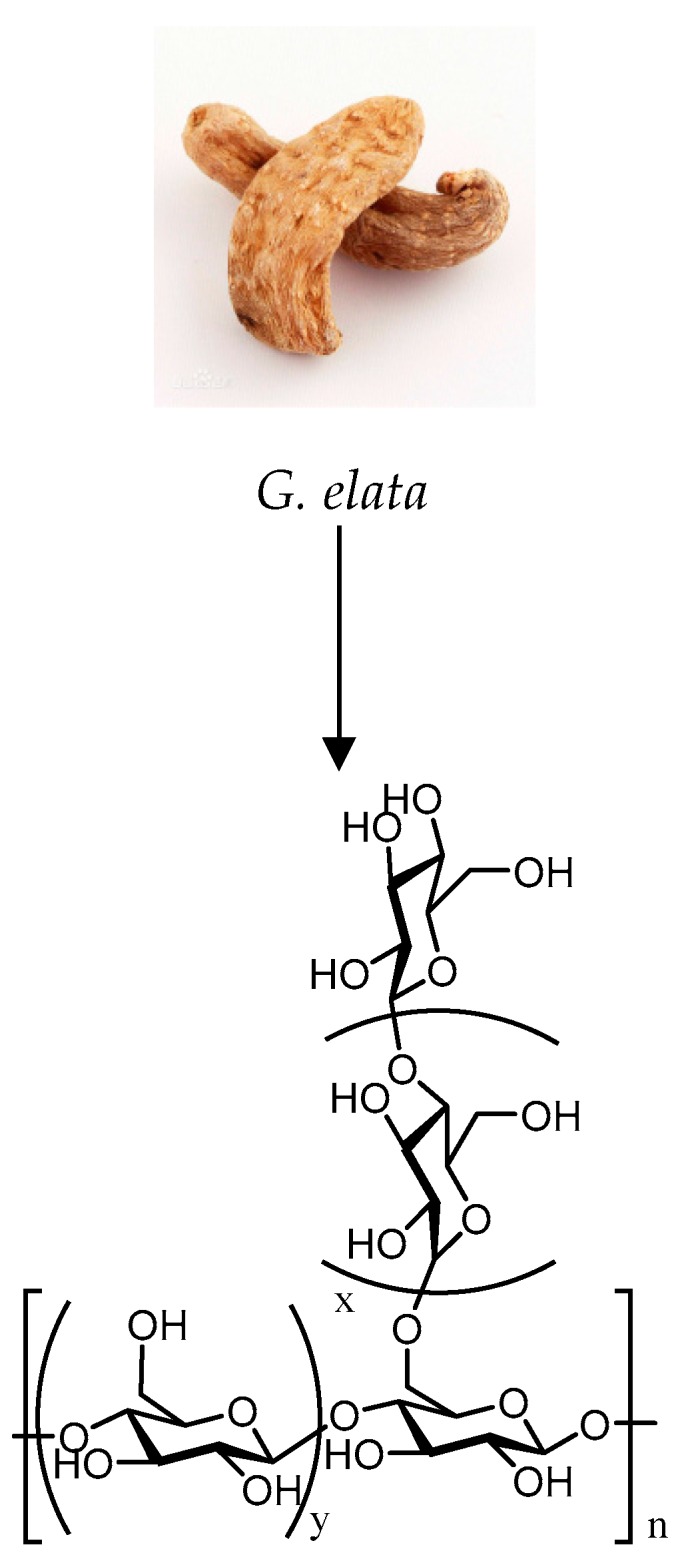
Chemical structure of part of the polysaccharides derived from *G. elata*: x + y = R; 1. AGEW: R = 14; 2. WTMA: R = 15; 3. WGEW: R = 16; and 4. PGEB-3*H*: R = 20 [5].

**Figure 2 molecules-24-02436-f002:**
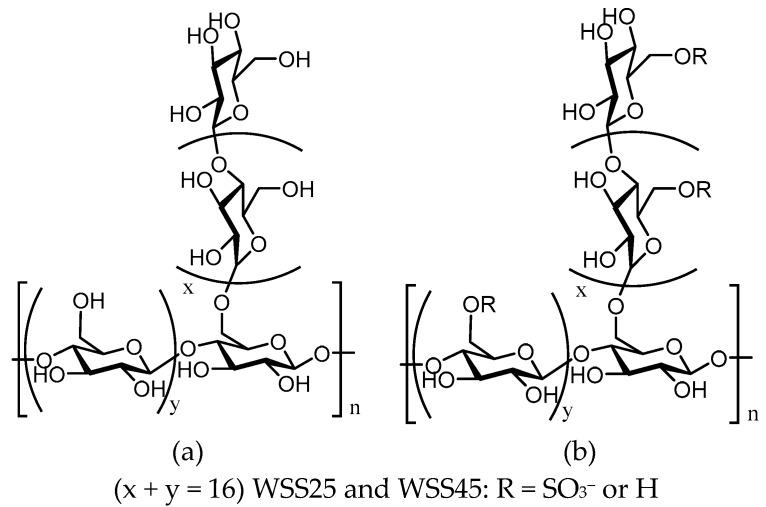
Structures of WGEW (**a**) and its sulfated derivatives WSS25 and WSS45 (**b**) [4].

**Figure 3 molecules-24-02436-f003:**
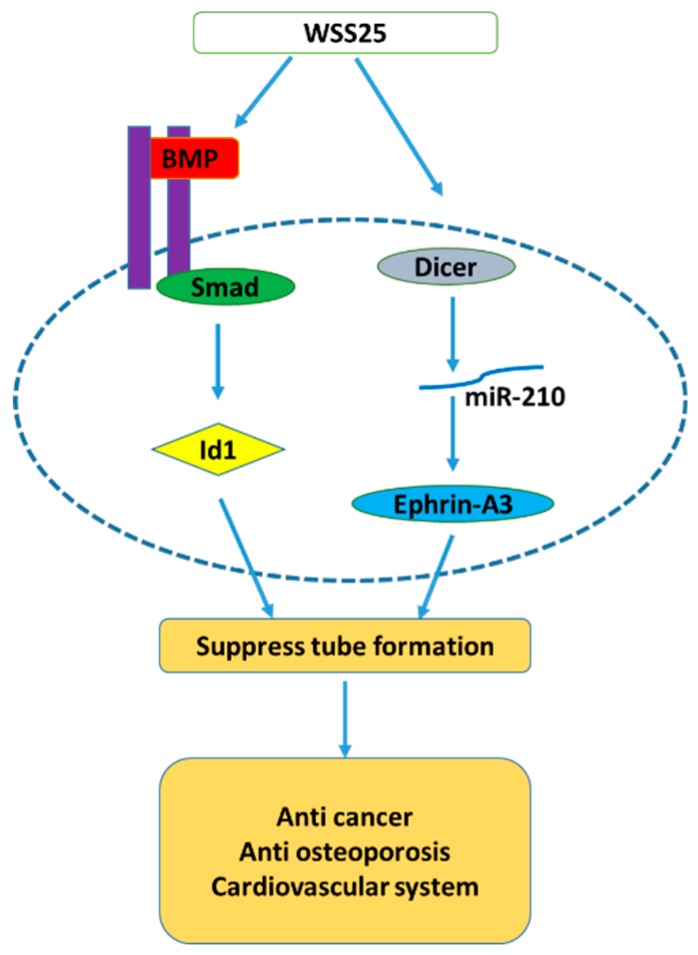
The pathway of tube formation suppression by the sulfated polysaccharides WSS25 from *G. elata*.

**Table 1 molecules-24-02436-t001:** The extraction methods of several polysaccharides from *G. elata*.

Name [ref]	Defat	Extract	Yield
WGE [10]	95% EtOH, 3 times	Boiling water, 4 times, 4 h	5.1 g
AGE [10]	95% EtOH, 3 times	5% NaOH (2 h) at 4 °C, 2 times	10 g
TM [27]	95% EtOH for 7 days	Boiling water, four times, 4 h	5.2%
PGEB-3H [26]	85% EtOH (1000 mL × 3) at 70 °C for 4 h	Water (800 mL × 4) at r.t. for 3 h	-
GR-0 [24]	10 volumes (v/w) of 80% EtOH at 60 °C, 2 times for 3 h	2 L of boiling distilled H_2_O for 3 h	2.47 g
RGP [23]	75% EtOH, 3 times over 24 h	Water at 74 °C, 3 times for 66 min	6.11%
GPs [13]	3 volumes of absolute EtOH at 60 °C for 24 h	Water at 90 °C, 4 times at 4 h each	-
The crude polysaccharide [28]	-	400 mL of distilled water for 2 h at 60 °C, 3 times	10.12%
GEP [54]	-	Water at 90 °C for 4 h	-

**Table 2 molecules-24-02436-t002:** The purification methods of several polysaccharides from *G. elata*.

Name [ref]	Purify	Flow Rate	Eluting Solvent	Yield
WGEW [10]	DEAE-cellulose column (50 × 5 cm)	-	deionized water	0.6 g from 5.1 g WGE
AGEW [10]	DEAE-cellulose column (50 × 5 cm)	-	deionized water	1.8 g from 10.0 g AGE
WTMA [27]	DEAE-cellulose (50 cm × 5 cm, Cl^−^ form)	-	0.1M NaCl	0.8 g from 6 g TM
PGEB-3H [26]	DEAE-cellulose A52 column (2.6 × 30 cm)	-	deionized water	0.797 g/kg
	Sephadex G-100 column (1.6 × 70 cm)	-	0.1 M NaCl	
The acidic polysaccharides [24]	DEAE-Sepharose CL-6B	-	0, 0.05, 0.1, 0.2, 0.3, 0.4, and 0.5 M NaCl	0.61 g from 2.47 g GR-0
RGP-1a, PGP-1b [23]	DEAE-cellulose-52 column (2.6 cm × 80 cm)	1 mL/min	deionized water	-
	Sephadex G-100 column (1.8 cm × 100 cm)	0.2 mL/min	deionized water	-
GPs [13]	DEAE-52 cellulose column	2.0 mL/min	distilled water and a gradient of 0→2 mol/L NaCl	-
PGE [28]	Sephadex G-200 column	0.30 mL/min	distilled water	-

**Table 3 molecules-24-02436-t003:** The molecular weight, backbone, monosaccharide composition, and biological activities of polysaccharides derived from *G. elata*.

Name [ref]	Molecular Weight (Da)	Monosaccharide Composition	Backbone	Biological Activities
WGEW [10]	1.0 × 10^5^	Glucose	α-1,4-glucan and α-1,4,6-glucan	-
AGEW [10]	2.8 × 10^5^	Glucose	α-1,4-glucan and α-1,4,6-glucan	-
WTMA [27]	7.0 × 10^5^	Glucose	α-1,4-glucan and α-1,4,6-glucan	Anti-cancer
PGEB-3*H* [25]	2.88 × 10^4^	Glucose	α-1,4-glucan and α-1,4,6-glucan	Cardiovascular system
The acidic polysaccharides [24]	-	Xylose, glucose, galacturonic acid, and glucuronic acid	-	Cardiovascular system
RGP-1a [23]	1.925 × 10^4^	fructose: glucose = 1:10.68	-	Immunological activity
RGP-1b [23]	3.92 × 10^3^	Glucose	-	Immunological activity
GPs [13]	2.71 × 10^5^	Glucose	-	Immunological activity
PGE [28]	1.54 × 10^6^	Glucose	α-1,4-glucan, α-1,3-glucan and α-1,4,6-glucan	Cardiovascular system
GEP [54]	875185	Glucose	-	Antioxidant

**Table 4 molecules-24-02436-t004:** Molecular weight, derivatization type, degree of substitution, and bioactivity of polysaccharides from *G. elata* and their derivatives.

Name [ref]	Molecular Weight (Da)	Modification	DS ^1^	Bioactivities ^2^		
				a	b	c
AGEW [10]	1.0 × 10^5^	-	0	−	−	−
ASS25 [10]	1.5 × 10^5^	sulfation	0.579	−	−	−
ASS45 [10]	6.8 × 10^4^		0.624	+	−	−
WGEW [10]	2.8 × 10^5^	-	0	−	−	−
WSS25 [10,11,95,96,97]	6.5 × 10^4^	sulfation	0.206	+	+	+
WSS45 [10]	1.9 × 10^5^		1.685	+	−	−
M1S [97]	1.8 × 10^5^		1.050	−	+	−
M2S [97]	1.3 × 10^5^		1.220	−	+	−
M3S [97]	7.5 × 10^4^		1.270	−	+	−
M4S [97]	4.1 × 10^4^		1.210	−	+	−
M5S [97]	1.4 × 10^4^		1.050	−	−	−
M6S [97]	1.2 × 10^4^		1.240	−	−	−
M7S [97]	2.7 × 10^3^		1.350	−	−	−
WGES1 [97]	2.4 × 10^5^		0.141	−	+	−
WGES2 [97]	6.7 × 10^4^		0.097	−	+	−
WGES3 [97]	1.8 × 10^5^		0.194	−	+	−
WGES4 [97]	8.0 × 10^4^		0.173	−	+	−
WGES5 [97]	1.38 × 10^5^		0.220	−	+	−
WGES6 [97]	7.6 × 104		0.202	−	+	−
WGEA	5.7 × 10^3^	aminopropylation	unknown	−	−	−
WGEC	7.3 × 10^4^	carboxymethylation		−	−	−
WGEP	2.2 × 10^3^	phosphorylation		−	−	−
WGEL	6.0 × 10^5^	acetylation		−	−	−

^1^ DS is calculated as 162 × %W/(96-80 × %W); %W is the content of SO_4_
^2−^. ^2^ a. Anti-dengue virus; b. antiangiogenesis; c. antiosteoporosis.

**Table 5 molecules-24-02436-t005:** Pharmacological activities information about polysaccharides isolated from *G. elata*.

Name [ref]	Activities	Cell Lines	Animals Model	Model of Action
WTMA [27]	anti-cancer	PANC-1, live LO2 cells	-	-
*G. elata* polysaccharides [9]		-	H22 tumor-bearing mice	increases caspase-3,8,9 levels and G0/G1 phase cell percentage, and decrease G2/M phase cell percentage
*G. elata* polysaccharides [12]	anti-aging and antioxidation	-	aging mice	improves the activities of SOD and GSH-Px, inhibits MAO activity, and reduces the level of MDA to
*G. elata* polysaccharides [102]		-	aging mice	promote the recovery of cranial nerves, significantly improve the activity of enzymes related to oxidative metabolism in the body
GEP [52]		-	aging mice	increases the activity of superoxide dismutase and glutathione peroxidase, as well as the serum and malondialdehyde levels
RGP-1a, RGP-1b [23]	Immunomodulatory effects	RAW 264.7 cell macrophages	-	enhances NO production and phagocytic activity
*G. elata* polysaccharides [103]		-	Immunocompromised mice	increases serum IgA, IgG, and hemolysin, the spleen index, thymus index, and serum IgM levels
GPs [13]		-	Kunming mice	augments serum IL-2, TNF-α, IFN-γ, IgG, IgA, and IgM levels, as well as the spleen and thymus indexes
PGEB-3*H* [25]		-	mice	increases the Ach content in brain tissue
PGB [104]		BDNF-positive cells and SCF-positive cells	Focal cerebral ischemia rats	up-regulates BDNF, Nestin, and SCF expression
GEP [105]	neuroprotection cardiovascular system	PC12 cells	-	inhibits the endoplasmic reticulum stress-mediated pathway
PGB [104]		-	RHR rats	promotes the production of endogenous vasoactive substances such as nitric oxide and inhibits the release of endogenous vasoconstrictors such as plasma endothelin and angiotensin II
PGEB-3*H* [25]		-	Hyperlipidemia rats	-
Crude and acidic polysaccharides [106]	cardiovascular system	-	SD rats fed a high-fat diet	suppresses total cholesterol and LDL
Acidic polysaccharides [24]		-	SHR rats fed a high-fat diet	reduces total cholesterol, triglyceride, and LDL levels
PGE [28]		-	-	-

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
