# Peer review of "Gastrodia elata Blume Polysaccharides: A Review of Their Acquisition, Analysis, Modification, and Pharmacological Activities"

_molecules, 2019, doi:10.3390/molecules24132436_

Reviewer 1 Report

This manuscript needs extensive English editing indeed. It reads like this species is not important any longer: "Gastrodia elata was a precious traditional Chinese herbal medicine...". Also, I don't think citations are in order in the abstract.

I suggest an extensive review by the author themselves before re-submitting. 

Author Response

We have checked the manuscript carefully, and a lot of errors were found. All the errors were revised. Furthermore, many contents was rewrote in manuscript.

Reviewer 2 Report

The manuscript compiles some potentially valuable information on the polysaccharides of Gastrodia elata and can thus be considered for publication. However, several aspects of its contents should be revised and complemented in order to make the article more useful for the reader.

In lines 52-57, the authors describe that different ethanol concentrations have been used for degreasing the material. Instead of simply stating the different concentrations, it would be much more useful for the reader if the authors could comment on whether the used ethanol concentration affects the outcome and in case yes, which concentration seems to be the most advisable one.

In lines 72-73, a numeric valued for extraction yield is given foa a method applying a specific enzymatic treatment. This is not informative at all for the reader since there is nothing that this value could be compared to. It is recommended to add some numeric information on extraction yields with other procedures as well to enable any conclusions on the relative performance of different methods.

In lines 99-100, it is written

 The merits of these methods are compatible to polysaccharide.

Please clarify. What does this mean?

Table 1 is rather bulky and simply lists the experimental parameters used by different studies. It is highly recommended to add some information on the extraction / purification outcome (yield) to add value for the reader. Also, splitting the table into two might help in keeping the table size smaller.

In section 3 describing the analysis methods, it is a bit difficult to follow which of the listed methods have actually been used for analyzing Gastrodiaelata and which of them are methods suggested for use based on results with other plant material. This section should be revised in order to clarify this aspect.

In line 257, “inhibition strength” should be replaced with an appropriate pharmacological term. Does this refer to potency or efficacy?

 In sections describing the biological activities, it is necessary to consider the pharmacokinetic properties of the polysaccharides in systemic and cellular level. Some of the proposed modes of action seem to require that the polysaccharides could actually enter a mammalian cell, which is not trivial for such huge and polar molecules. Whether the polysaccharides are able to enter cells and by which mechanism should be discussed in the text. Regarding the systemic level kinetics, the route of administration applied in animal studies should be mentioned, as well as pharmacokinetic parameters such as oral bioavailability, serum halflife and potential means of metabolism and excretion.

Author Response

Comment 1:

In lines 52-57, the authors describe that different ethanol concentrations have been used for degreasing the material. Instead of simply stating the different concentrations, it would be much more useful for the reader if the authors could comment on whether the used ethanol concentration affects the outcome and in case yes, which concentration seems to be the most advisable one.

Answer:

As for the concentration of degreasing G.elata, it maybe effect the yield of G.elata. Chen et al. obtained crude G.elata polysaccharide RGP degreased with 75% ethanol with a yield of 6.11% [23]. Lee et al. obtained 2.47 g the crude G. elata polysaccharide degreased with 80% ethanol [24]. Ming et al. found the yield of G. elata polysaccharide PGEB-3H degreased with 85% ethanol was 0.797g/kg [25]. Chen et al. found the yield of crude G. elata polysaccharide TM degreased with 95% ethanol was 5.2% [27]. The yield is related to many factors, such as extraction time, extraction temperature, extraction frequency and so on. It is difficult to judge the best degreased concentration without the single factor study.

Comment 2:

In lines 72-73, a numeric valued for extraction yield is given for a method applying a specific enzymatic treatment. This is not informative at all for the reader since there is nothing that this value could be compared to. It is recommended to add some numeric information on extraction yields with other procedures as well to enable any conclusions on the relative performance of different methods.

Answer:

I added some information about the yield of extraction method in this manuscript. Zhu et al. found the yield of G. elata polysaccharides was 5.42% with microwave extraction [30]. Zhang et al. found the yield of G. elata polysaccharides was 32.78% with ultrasound extraction [35]. Tan et al. found the extraction amount of G. elata polysaccharides was 46.63 mg/g with enzyme extraction [38]. Wang et al. showed that the yield of G.elata polysaccharide extraction by hot water extraction was 22.380%; the yield of G.elata polysaccharides by ultrasonic assisted extraction was 33.089% and enzymatic extraction of G.elata polysaccharide yield was 50.315% [40]. On the whole, the yield of G.elata polysaccharide with enzymatic extraction is higher than other methods.

Comment 3:

In lines 99-100, it is written" The merits of these methods are compatible to polysaccharide." Please clarify. What does this mean?

Answer:

What I want to mean is that these methods are all suitable for deproteinization of polysaccharide solutions.

Comment 4:

Table 1 is rather bulky and simply lists the experimental parameters used by different study. It is highly recommended to add some information on the extraction / purification outcome (yield) to add value for the reader. Also, splitting the table into two might help in keeping the table size smaller.

Answer:

I added some information on the extraction / purification outcome in the table. At the same time, the table 1 was splited into two.

Comment 5:

In section 3 describing the analysis methods, it is a bit difficult to follow which of the listed methods have actually been used for analyzing Gastrodia elata and which of them are methods suggested for use based on results with other plant material. This section should be revised in order to clarify this aspect.

Answer:

I revised the third part. Matrix-assisted laser desorption/ionization,high performance thin layer chromatography, high performance liquid chromatography, capillary electrophoresis tubes and PACE are suggested to apply for analyzing G. elata polysaccharides according to other plant material. The methods described in the last paragraph of section 3.2 are also suggested based on results from other plant materials.

Comment 6:

In line 257, “inhibition strength” should be replaced with an appropriate pharmacological term. Does this refer to potency or efficacy?

Answer:

The description herein is intended to express the effect of changes in the properties of the derivatized polysaccharide on the final inhibitory effect. “Inhibition strength” was replaced by “efficacy”.

Comment 7:

In sections describing the biological activities, it is necessary to consider the pharmacokinetic properties of the polysaccharides in systemic and cellular level. Some of the proposed modes of action seem to require that the polysaccharides could actually enter a mammalian cell, which is not trivial for such huge and polar molecules. Whether the polysaccharides are able to enter cells and by which mechanism should be discussed in the text. Regarding the systemic level kinetics, the route of administration applied in animal study should be mentioned, as well as pharmacokinetic parameters such as oral bioavailability, serum halflife and potential means of metabolism and excretion.

Answer:

I carefully read again all the references cited on the section of pharmacologically activity of Gastrodia elata polysaccharide. And I found that these researches on the pharmacological activities of G. elata polysaccharides mentioned in this paper mainly involve two parts: in vivo animal experiment and in vitro cell experiment. In addition to Liu et al. used the intracortical injection as administration method in the experiment [12], almost all animal experiments were performed by oral administration. However, none of the above study on systemic activity have mentioned any pharmacokinetic parameters and characteristics other than the mode of administration and dosage of G. elata polysaccharides. And few of study have mentioned whether G. elata polysaccharides can enter cells or through other means affect cells. I also added related expression in this review.

Reviewer 3 Report

The review of Zhu et al. on Gastrodia elata polysaccharides is a wide description of occurrence, method of extraction and purification and use of these natural product.

Actually, the review cover a topic of minor interest in medicine, but the comprehensive description of chemical features of Gastrodia elata polysaccharides is intersting. 

Extensive editing of English language and style is mandatory for the manuscript being not acceptable in this form.

The botanical name must be written in the title and the first time as: Gastrodia elata Blume and then always G. elata.

Finally, I suggest to improve the discussion in order to better focus the perspective of use of this medicinal plant in human health. 

Author Response

Comment 1:

The botanical name must be written in the title and the first time as: Gastrodia elata Blume and then always G. elata.

Answer:

The botanical name of Gastrodia was uniformly expressed as G. elata.

Comment 2:

Finally, I suggest to improve the discussion in order to better focus the perspective of use of this medicinal plant in human health. 

Answer:

Finally, I added a statement about the potential of G. elata polysaccharides to improve the health of middle-aged and elderly people, in the section of discussion. It is undeniable that G. elata polysaccharides has good biological activity in many aspects, including anti-cancer, anti-oxidation, anti-hypertensive, immune system and nervous system regulation. Especially, the function of G. elata polysaccharides in neuroprotection and treatment of hypertension diseases has great potential to improve the situation health of the middle-aged and elderly population. However, most of the current study on systemic pharmacological activity have neglected the mechanism of the action of G. elata polysaccharide in animal experiments in vivo. The molecular mechanisms in in vitro cell experiments have also not been fully investigated. The pharmacokinetic study of G. elata polysaccharides is still blank.

Round  2

Reviewer 1 Report

About the manuscript "Gastrodia elata Bl Polysaccharides: a Review of Acquiring, Analysis, Modification and Pharmacological Activities", again I suggest an intensive English language editing. It is still Unreadable and it could be more organized.

Author Response

We read our article carefully and corrected some mistakes in it.

Reviewer 3 Report

Authors revised and improved the manuscript according to reviewers' suggestion,

Nevertheless I truly suggest to consider an english revision made by an english native speaker.

Botanical name should be corrected in this way:

Line 2: Gastrodia elata Blume (Blume not in italic). The same in line 19 and line 29.

Always write G. elata and other latin names (such as in vitro and in vivo) in italic.

Author Response

(The authors gave the same response as above.)
